# A long noncoding RNA acts as a post-transcriptional regulator of heat shock protein (HSP70) synthesis in the cold hardy *Diamesa tonsa* under heat shock

Paola Bernabò[1,2], Gabriella Viero[2], Valeria Lencioni[1]*

1 Department of Invertebrate Zoology and Hydrobiology, MUSE-Museo delle Scienze, Trento, Italy,
2 Institute of Biophysics-CNR Trento Unit, Povo, Trento, Italy

* valeria.lencioni@muse.it

**Data Availability Statement:** All relevant data are within the manuscript and its Supporting Information files.

## Abstract

Cold stenothermal insects living in glacier-fed streams are stressed by temperature variations resulting from glacial retreat during global warming. The molecular aspects of insect response to environmental stresses remain largely unexplored. The aim of this study was to expand our knowledge of how a cold stenothermal organism controls gene expression at the transcriptional, translational, and protein level under warming conditions. Using the chironomid *Diamesa tonsa* as target species and a combination of RACE, qPCR, polysomal profiling, western blotting, and bioinformatics techniques, we discovered a new molecular pathway leading to previously overlooked adaptive strategies to stress. We obtained and characterized the complete cDNA sequences of three heat shock inducible 70 (*hsp70*) and two members of heat-shock cognate 70 (*hsc70*). Strikingly, we showed that a novel pseudo-hsp70 gene encoding a putative long noncoding RNA (lncRNA) which is transcribed during thermal stress, acting as a ribosome sponge to provide post-transcriptional control of HSP70 protein levels. The expression of the pseudo-hsp70 gene and its function suggest the existence of a new and unexpected mechanism to cope with thermal stress: lowering the pace of protein production to save energy and optimize resources for recovery.

## Introduction

Understanding how freshwater species potentially react and adapt to climate change is a major challenge in predicting future biodiversity trends [1]. This is particularly important in high mountain freshwaters where migration and dispersion to escape stressors are hampered by isolation and habitat fragmentation [2, 3, 4]. Owing to shrinking glaciers, water temperature of glacier-fed streams is increasing while their discharge is decreasing, affecting taxonomical and functional diversity of the animal and vegetal communities [5, 6]. Cold stenothermal species adapted to live at temperatures close to their physiological limits might only survive and reproduce if they can adapt to new environmental conditions or if they are able to avoid the stressor adopting specific behaviours [7]. Barring these abilities, they are expected to disappear [8]. In

**Funding:** This work was partially supported by the Cassa di Risparmio di Trento e Rovereto Foundation within the RACE-TN Project ("Valutazione del rischio ambientale dei contaminanti emergenti nei fiumi trentini: effetti sulla vita selvatica e sull'uomo"/"Environmental Risk assessment of emerging contaminants in Trentino rivers: effects on wildlife and human health", Grant CARITRO n. 2015.0199; October 2015-January 2018) coordinated by Valeria Lencioni. The funder had no role in study design, data collection and analysis, decision to publish, or preparation of the manuscript. The funder supported 1 m/m of the contract of the first author (Paola Bernabò) and the purchase of some consumables. There was no additional external funding received for this study.

**Competing interests:** The authors have declared that no competing interests exist.

the mountaintop environment exposure is magnified because the rate of warming is amplified with elevation, with high mountain ecosystems experiencing more rapid rate of temperature increase than those in lowlands [9]. Studying the adaptive potential of these species is indeed essential for prediction of the consequences on biota and habitats of global warming in glacierized regions [10].

To detail the molecular response to increased temperature in cold stenothermal species inhabiting glacier-fed streams, we chose *Diamesa tonsa* (Haliday), belonging to the *cinerella* species group (Diptera: Chironomidae). Six Palaearctic species are ascribed to this group, all inhabiting cold running waters, with *D. tonsa* being one of the most common in the Mediterranean Basin as well as Northern Europe and Russia [11]. Larvae of *D. cinerella* gr. are freeze-tolerant, with a thermal optimum below 6°C [12], and survive short-term heat shock (HS) by developing a Heat Shock Response (HSR) based on the synthesis of Heat Shock Proteins (HSPs) [13].

HSR based on HSPs has been characterized for a wide range of species and found to exhibit a high degree of conservation of its basic properties across prokaryotes and eukaryotes [14]. Under non-stressful conditions HSPs act as molecular chaperones to stabilize actively denaturing proteins, refold proteins that have already denatured, and direct irreversibly denatured proteins to the proteolytic machinery of the cell [15, 16, 17]. Also, under non-stressful conditions HSPs facilitate the correct folding of proteins during translation and their transport across membranes [18, 19]. Among HSPs the 70 kDa family, consisting of inducible (HSP70) and constitutive (heat shock cognate, HSC70) forms, is the most studied in relation to thermal stress and has been found in all organisms investigated to date [20, 21]. [22] and [13] demonstrated that HSC70 plays a role in cold resistance for *D. cinerella* gr. larvae and that the HSR is comprised of a strong transcriptional boost of the inducible HSP70 gene at a temperature six times higher than that at which they live in nature. The involvement of HSC70 and HSP70 in cold and heat tolerance was observed in other cold adapted chironomids such as adults of *Belgica antarctica* [23] and larvae of *Pseudodiamesa branickii* [24]. Knowledge of how these insects control gene expression at the transcriptional, translational and protein level under heating is still under-explored. The present study aims to address this using a multi-level approach to study HSR at the transcriptional, translational and protein level in *D. tonsa*. This is particularly important because studying changes in gene expression only at the transcriptional level may be misleading [25, 26] given the poor average correlation between protein and transcript [27, 28] due to post-transcriptional controls of gene expression. Here we hypothesize that, similar to observations in higher eukaryotes, post- transcriptional control of *hsp70* and *hsc70* gene expression in insects may exist. In line with this hypothesis, we identified a novel bio-molecular process in cold adapted organisms, shedding light into a new adaptation strategy to cope with heat stress involving a long noncoding RNA (lncRNA). Intriguingly, over the last few years, lncRNA with putative regulatory function relating to HSR has begun to attract attention, and several of lncRNAs are implicated in mammalian HSR [29]. At present, there are no reports about the involvement of lncRNAs in organisms that, like *D. tonsa*, are actually facing global warming challenges in the wild.

## Material and methods

### Animal model and collection

Fourth-instar larvae of *D. tonsa* were used as an animal model. *D. tonsa* is a Palaearctic species well distributed in the European mountain regions, particularly frequent in the Alps and the Apennines (Fauna Europaea: https://fauna-eu.org/). Larvae were collected with a 30 × 30 cm pond net (100 μm mesh size) in mats of the chrysophyte *Hydrurus foetidus*, in winter 2016, in

the glacier-fed stream Frigidolfo (Lombardy Province, NE Italy, 1584 m a.s.l., 10°30′19.32″ N; 46°17′ 51.07″ E). The Frigidolfo stream was characterized by clear (3.8±1.4 Nephelometric Turbidity Unit) and well oxygenated (per cent oxygen saturation = 80% - 90%) waters, with a mean temperature of 4°C during the sampling period, recorded using a field multiprobe (Hydrolab Quanta, Hydrolab Corporation®, Texas, USA). Larvae were sorted in the field with tweezers, transferred to plastic bottles filled with stream water and transported to the laboratory via cooling bag within 2 hours of collection. Animals were reared in 500-mL glass beaker (max 50 specimens/beaker) with filtered stream water (on Whatman GF/C, particle retention 1.2 μm) in a thermostatic chamber (ISCO, mod. FTD250-plus) at 4°C with an aerator to maintain oxygen saturation above 80%. The rearing temperature (4°C) corresponded to the mean environmental temperature during sampling period [24].

Species identification was confirmed by head capsule morphology observed under the stereomicroscope (50 X) according to [30] within 24 h of sampling and by DNA Barcoding analysis of a sub-sample of the collected larvae based on mitochondrial COX1 gene sequence [11].

## Heat shock exposure

Larvae of *D. tonsa* were exposed for 1 h to three different stress temperatures (15°C, 26°C and 32°C), chosen according to [13]: 26°C is the highest temperature at which all the tested larvae were found alive after 1 h of stress; 32°C is the $LT_{50}$ of the larvae in winter season; 15°C selected as an intermediate temperature between the natural-ideal and sub-lethal temperatures.

3–5 groups of 5 larvae each were transferred to 25-mL plastic bottles (Kartell, Italy) filled with 10 mL of preheated filtered stream water, under aeration to avoid mortality due to oxygen depletion. Larvae were then maintained at the stress temperature for 1 h. In all, 3–5 replicates of 5 specimens each were maintained at 4°C for the entire period of each treatment and used as control (Ctrl). After treatments, the larvae were returned to 4°C (= rearing temperature) for 1 h and then only living (moving spontaneously) larvae were immediately flash frozen in liquid nitrogen. The larvae were stored at -80°C until further analyses.

## Isolation of total mRNA and reverse transcription

Total RNA was extracted from 2–5 larvae using a commercial kit (TRIZOL, Life Technologies, Carlsbad, CA, USA), according to the manufacturer's protocol. RNA concentration was determined by UV absorption using a NanoDrop ND-1000 spectrophotometer (Thermo Fisher Scientific, Waltham, MA, USA) and quality checked by agarose gel electrophoresis. Total RNA (1 μg) was then reverse transcribed using the First Strand cDNA Synthesis Kit (Thermo Fisher Scientific) and oligo(dT) as primers.

## hsp70 amplification

cDNA (1 μL) from control larvae was amplified using the KAPA HiFi HotStart DNA Polymerase Taq (Kapa Biosystems, Wilmington, MA, USA), insect-HSP70 degenerate primers (from Bernabò *et al.*, 2011; Table 1) with a touchdown PCR protocol(annealing from 61 to 50°C, - 1°C/cycle + 30 cycles at 50°C).

PCR product was observed using agarose gel electrophoresis, cloned using the CloneJet PCR Cloning kit (Thermo Fisher Scientific), and transformed into DH5α competent cells. A minimum of 40 colonies were analysed for the presence, size, and sequence of the PCR insert using colony PCR-RFLP analysis (TRu1 l restriction enzyme). Three different electrophoresis restriction patterns were observed, and the corresponding inserts were sequenced.

**Table 1. List of primers (5′-3′) used to amplify *hsp70* with and without intron, *hsc70-Isfgsgs*, *hsc70-II*, and *actin* in RT-PCR analysis.**

| Primer | Application | Sequence (5′– 3′) |
|---|---|---|
| Deg_Hsp70-F | 5′-3′ RACE PCR | ACVGNTCCNGCNTAYTTYAAYGA |
| Deg_Hsp70-R | 5′-3′ RACE PCR | GCNACNGCYTCRTCNGGRTT |
| GSP-HSP70_F | 5′ RACE PCR | TGATGCAAAGCGGCTGATTGGACGTA |
| GSP-HSp70_R | 3′ RACE PCR | TTGGCAGCATCTCCAATTAATCTTTCTGTATC |
| GSP-HSC70I_F | 5′ RACE PCR | GTCGTAAATTCGATGACCCC |
| GSP-HSC70I_F HSC70I_R | 3′ RACE PCR | GACATTACGTTCTCCAGCAG |
| GSP-HSC70II_F | 5′ RACE PCR | GTTTGATCGGTCGTGAATGGAG |
| GSP-HSC70II_R | 3′ RACE PCR | CCCAAATGTGTGTCACCGTTT |
| Hsp70 Forward | Expression (Fig 2A) | TTGGGAACAACATATTCCTGC |
| Hsp70 Reverse | Expression (Fig 2A) | TTCGTTTAGCATCAAAGACACTG |
| Hsc70I Forward | qPCR (Fig 3B and 3C) | GTCGTAAATTCGATGACCCC |
| Hsc70I Reverse | qPCR (Fig 3B and 3C) | GACATTACGTTCTCCAGCAG |
| Hsc70II Forward | qPCR (Fig 3B and 3C) | GTTTGATCGGTCGTGAATGGAG |
| Hsc70II Reverse | qPCR (Fig 3B and 3C) | CCCAAATGTGTGTCACCGTTT |
| Hsp70 tot F | qPCR (Fig 3B and 3C; Fig 4A, 4B and 4C) | TGTTGGAGTTTATCAACATGGA |
| Hsp70 tot R | qPCR (Fig 3B and 3C; Fig 4A, 4B and 4C) | TTTGGCAGCATCTCCAATTA |
| Hsp70 genomic F | gDNA amplification (Fig 2B and 2C) | GAAACAGAACAACACCCAGCT |
| Hsp70 genomic R | gDNA amplification (Fig 2B and 2C) | ACTTCAGCAGTTTCACGCAT |
| Actin Forward | qPCR (Fig 3B and 3C; Fig 4C and 4D, 4E and 4F) | CTGCCTCAACCTCATTGGAAAA |
| Actin Reverse | qPCR (Fig 3B and 3C; Fig 4A, 4B and 4C) | TGGTTGTAGACGGTTTCGTG |

## 5′ and 3′ RACE PCR

Amplification of 5′ and 3′ ends of cDNA was performed according to the protocol described in the SMART RACE cDNA Amplification Kit (Takara Bio USA, Inc. Mountain View, CA, USA). Three pairs of gene-specific primers (GSP), one for each hsp70 isoform: *hsp70*, *hsc70-I* and *hsc70-II*, were designed based on the sequences obtained from amplification with degenerate primers (Table 1).

Total RNA of *Diamesa tonsa* larvae was extracted according to the Trizol protocol (Thermo Fisher Scientific) and the quantity and quality assessed as described above. 1 μg of total RNA was first retrotranscribed with primers supplied in the kit and this first-strand cDNA was used directly in PCR amplification reactions that were achieved using a high-fidelity enzyme (KAPA HiFi DNA Polymerase, Kapa Biosystems), the Universal Primer Short (UPS, supplied by the kit), and gene-specific primer (GSP_F for 3′ RACE- cDNA or GSP_R for 5′ RACE- cDNA). The PCR was performed with the following cycler protocol: 95°C for 3 min, 25 cycles of 98°C for 30 sec, 68°C for 15 sec and 72°C for 90 sec, and a final extension of 5 min at 72°C.

## Cloning and sequencing of 5′ and 3′-RACE PCR products

PCR products were run and purified by agarose gel electrophoresis and cloned into the pJET 1.2 cloning vector (Thermo Fisher Scientific). The inserts were sequenced.

## Sequences analysis

Full-length cDNA sequences have been deposited in GenBank under accession numbers KC860254 (*Dc-hsp70*), KC860255 (*Dc-hsc70-I*) and KC860256 (*Dc-hsc70-II*). When sequences were deposited, they were ascribed to a *Diamesa cinerella* gr., referred to as *Dc-*, only successively was the species identification confirmed as *D. tonsa*. Throughout the paper we decided to refer the genes to the species "tonsa" as *Dt-*. The sequences were used to search for homology in other organisms by BLAST software on the NCBI website (https://blast.ncbi.nlm.nih.gov/Blast.cgi).

Sequence alignments were carried out using the Bioedit software package and ORFs were identified with the aid of the software ORF Finder (http://www.ncbi.nlm.nih.gov/projects/gorf/). Molecular weights of the predicted proteins were calculated by Compute pI/Mw tool (ExPASy) (http://web.expasy.org/compute_pi/). The phylogenetic trees were constructed using the "One Click" mode with default settings in the Phylogeny.fr platform (http://www.phylogeny.fr) [31].

The presence of possible splicing sites was predicted using the BDGP: Splice Site Prediction by Neural Network (http://www.fruitfly.org/seq_tools/splice.html).

Specific primers for expression analysis were designed from the full-length cDNA sequences of *Dt-hsp70*, *Dt-hsc70I* and *Dt-hsc70II* (Table 1) using Primer3Web (http://primer3.ut.ee) and OligoCalc (http://biotools.nubic.northwestern.edu/OligoCalc.html).

## Polysomal extraction

Polysomes were extracted as reported in [26]. Briefly, 5 frozen larvae were pulverized under liquid nitrogen in a mortar with a pestle and the powder lysed in 0.8 mL of lysis buffer (10 mM Tris-HCl at pH 7.5, 10 mM NaCl, 10 mM MgCl$_2$, 1% Triton-X100, 1% Na- deoxycholate, 0.4 U/μL SUPERase. In RNase Inhibitor (Life Technologies), 1 mM DTT, 0.2 mg/mL cycloheximide, 5 U/mL Dnase I). Following lysis, cellular debris was removed by centrifugation (13.200 rpm, 2 min at 4˚C) and the supernatant was kept ice-cold for 15 minutes. The cleared lysate was then centrifuged at 13.200 rpm, 5 min at 4˚C to remove all nuclei and mitochondria. The supernatant obtained was layered onto a 12 mL linear sucrose gradient (15%–50% sucrose [w/v], in 30 mM Tris-HCl at pH 7.5, 100 mM NaCl, 10 mM MgCl$_2$) and centrifuged in a SW41Ti rotor (Beckman) at 4˚C and 197.000 *g* for 100 min in a Beckman Optima LE-80K Ultracentrifuge. 1 mL fractions were collected with continuous absorbance monitoring at 254 nm using an ISCO UA-6 UV detector.

## Polysomal RNA extraction

Sucrose fractions from the entire polysomal profile were divided into two groups (subpolysomal and polysomal fractions) and the pooled fractions were treated with 1% SDS and proteinase K (100 μg/mL) for 75 minutes at 37˚C before phenol–chloroform RNA extraction. Polysomal RNA pellet was resuspended in 25 μL of Rnase-free water. RNA was quantified by Nanodrop and the quality was assessed by agarose gel electrophoresis.

## Quantitative real-time RT-PCR (qPCR)

Subpolysomal and polysomal RNA (500 ng) was reverse transcribed using the First Strand cDNA Synthesis Kit (Thermo Fisher Scientific) following the manufacturer's instructions. cDNA was amplified by Real-Time PCR using Kapa Sybr Fast qPCR Mastermix (Kapa Biosystems) and specific primers (Table 1) on a CFX96 Touch™ Real-Time PCR Detection System (Biorad, Hercules, CA, USA). *Actin* was used as housekeeping gene normalization control (Table 1). For each condition, 3 biological replicates were prepared, and for each biological replicate, 3–4 Real-Time PCR amplifications were run. Each primer pair was validated for dimer formation by melting curve analysis. Amplification profiles were analysed using CFX Manager Software (analysis of the melting Curve for the presence of primer dimer) and relative expression levels were calculated using the delta/delta Ct method [32]. The log2 ΔTE (change of Translation Efficiency) was calculated as the ratio between the fold change at the polysomal level and the fold change at the sub-polysomal level of the gene of interest.

## Protein extraction and Western Blot analysis

Protein was extracted from 5 live larvae using the methanol/chloroform protocol [33] and solubilized in electrophoresis sample buffer (Santa Cruz Biotechnology, Dallas, TX, USA) for SDS-polyacrylamide gel electrophoresis and Western Blot analysis. Gel electrophoresis was performed using 12% Mini-PROTEAN-TGX Precast Protein Gels (Bio-Rad) and transferred to nitrocellulose membrane. Immunoblotting was performed using GAPDH (1:1,000, Santa Cruz Biotechnology) and HSP70 (1:800, Abcam, Cambridge, UK) primary antibodies and the corresponding HRP-conjugated secondary antibodies (1:2,000, Santa Cruz Biotechnology). The blots were developed with SuperSignal™ West Femto Maximum Sensitivity Substrate (Thermo Fisher Scientific) and acquired on a ChemDoc-It (BioRad Laboratories). The image analysis was performed using the ImageJ (https://imagej.nih.gov/ij/) image processing package.

## Genomic DNA extraction and amplification

Genomic DNA was extracted from a single larvae of *D. tonsa* using the DNeasy Blood & Tissue Kit (Qiagen, Hilden, Germany), following the manufacturer's instructions. The gDNA concentration was estimated by Qubit and about 20 ng of gDNA amplified with primers specifically designed to include the intron region (Hsp70 genomic F/R, Table 1) using PuReTaq Ready-To-Go™ PCR Beads (GE Healthcare, Chicago, IL, USA).

## Gene copy number analysis

The relative copy number for *hsp70* with and without intron was estimated by Real-Time PCR analysis using SYBR Green dye. Two pairs of primers that specifically recognize *hsp70 with intron* (Intron-F and Intron-R, Table 1) or *hsp70 without intron* (Hsp70 no intron-F and Hsp70 no intron-R, Table 1) were used to amplify gDNA by Real-Time PCR using qPCRBIO SyGreen Mix Separate-ROX (PCR Biosystem, Wayne, PA, USA) with Mic qPCR Cycler (Bio Molecular System, Upper Coomera, Australia). Consistent efficiency of the two primer pairs was analysed by the amplification of a serial dilution of gDNA. Four different larvae gDNAs were amplified and the ΔCt between the *hsp70* and the *hsp70 + intron* signal calculated. The Relative Gene Copy Number was finally calculated using $2^{\Delta Ct}$.

# Results

## Cloning, sequencing, and characterization of the Dt-hsp70 transcripts

For detailed study of the HSR of *Diamesa tonsa*, we produced the complete sequences of *hsp* transcripts in this organism. We cloned the *hsp* transcripts, using degenerate primers based on a conserved region of insect's *hsp70*, and performed RACE PCR to amplify cDNAs. With this approach we successfully obtained sequences for three unique transcripts (Fig 1A and S1 and S2 Figs). The nucleotide sequence of the first transcript was highly homologous to the well-known *hsp70* inducible isoform (Fig 1A, *Dt-hsp70*). The remaining transcripts were likely the constitutive isoforms (S1 and S2 Figs, *Dt-hsc70-I*, *Dt-hsc70-II)*.

Upon performing a cross-species alignment, we found that the cDNA sequence of *Dt-hsp70* shared high homology with other inducible *hsp70s* from other insects: 81% with *Chironomus yoshimatsui* (AB162946.1) and 72% with *Phenacoccus solenopsis* (KM221884.1), strongly suggesting that this transcript indeed belongs to the *hsp70* family. The *Dt-hsc70-I* showed high homology with constitutive form *hsp70* from other Diptera: 82% with both *Sitodiplosis mosellana* (KM014659.1) and *Polypedilum vanderplanki* (HM589530.1). Similarly, we found that *Dt-hsc70-II* shared the highest homology with *P. vanderplanki* (HM589531.1) and *Acyrthosiphon pisum* (NM_001162948.1).

The alignment of deduced amino acid protein sequences resulted in characteristic hsp70 family motifs, further supporting our hypothesis that these transcripts are *bona fide* products of the hsp70 gene (Fig 1A and S1A Fig). In particular, we found the motifs IDLGTTYS, DLGGGTFD, and IVLVGG and the cytoplasmic *hsp70* carboxyl-terminal region (EEVD) in the case of *Dt-hsp70* and *Dt-hsc70-I*. In *Dt-hsc70-II* we also found the characteristic ER localization signal (KDEL) at the C-terminus (Fig 1A and S1A Fig). A summary of the characteristics of all sequences and their deduced proteins is displayed in Table 2.

Next, using both the nucleotide and the amino acid sequence of *Dt-hsp70*, we obtained two phylogenetic trees (Fig 1B and 1C). Interestingly, *Dt-hsp70* clustered with the *hsp70* of the closely related chironomids (*Chironomus* spp). The relationships displayed in the trees were in agreement with established Diptera phylogeny, with chironomids separated from other Nematocera and Brachycera [34, 35]. Similar results were obtained for *Dt-hsc70-I* and *Dt-hsc70-II* using the Phylogeny.fr tool on the ExPASy Proteomics server [31] (S2 Fig).

## Identification of a novel hsp70-pseudogene encoding a putative lncRNA

While analysing the response of the *Dt-hsp70* transcript to heat stress for 1 h and at 15˚, 26˚, and 32˚C and using primers for *Dt-hsp70*, we serendipitously observed that two amplicons were produced from cDNA amplification (Fig 2A and Table 1). The unexpected second PCR amplicon was about 400 bp longer than the expected one and was observed exclusively in larvae exposed to heat stress. Intrigued by these results, we cloned and sequenced this alternative transcript and compared it to *Dt-hsp70* (S3 Fig). This alternative transcript has a short intron of 404 bases (from nucleotide 421 to 854, donor site from nucleotide 421 to 435 [*prediction score* = 0.96]; acceptor score from nucleotide 835 to 875 [*prediction score* = 0.98]) and a polyA tail following nucleotide 1239. In human, hsp70 pseudogenes can act as lncRNAs. Thus, we suspected this novel transcript to have similar characteristics. To test this hypothesis, we used a computational prediction tool [36] to calculate the probability of our novel transcript being a lncRNA. As control we used the same tool on the *Dt-hsp70* transcript and obtained that *Dt-hsp70* is classified as a coding transcript with coding probability 1, Fickett score 0.41216, complete putative ORF of 635 aa in length, and a pI 5.67, in agreement with data shown in Table 2. Strikingly, the novel Hsp70 transcript was classified as a noncoding sequence with coding probability 0.300294, Fickett score 0.36989, a complete putative ORF of 110 aa, and a pI 8.70.

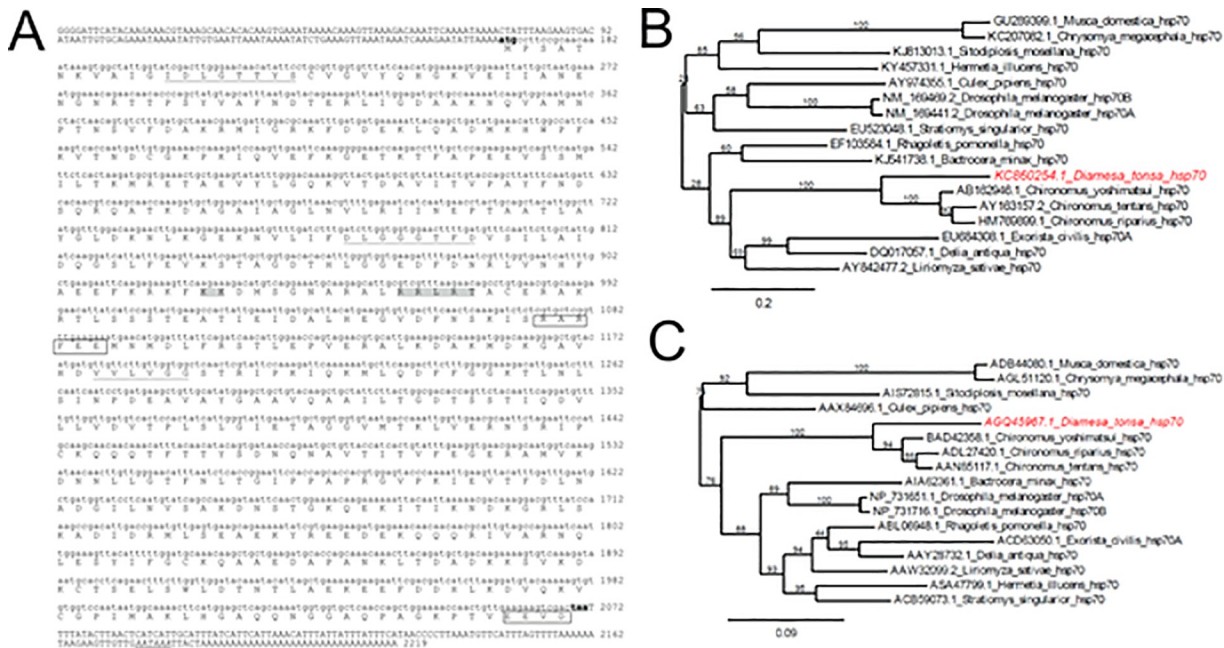

**Fig 1. Characterization of *hsp70* gene in *Diamesa tonsa*.** (**A**) Nucleotide sequence of the *hsp70* gene in *D. tonsa* with deduced amino acid sequence. In the nucleotide sequence, upper case indicates the 5′UTR and the 3′UTRs, whilst lower case indicates the coding region. The start codon (ATG) and stop codon (TAA) are shadowed and in bold, and the consensus polyA signal in the 3′UTR is in italic and double-underlined. The three characteristic signatures of the HSP70 family are underlined: the non-organelle consensus-motif (RARFEEL) and the cytoplasmic C-terminal region EEVD are shown. The putative bipartite nuclear localization signal (KK and RRLRT) is shadowed in grey. (**B**) Phylogenetic tree inferred from nucleotide sequences of *hsp70* in different dipteran species. The tree was constructed using Phylogeny.fr tool at ExPASy Proteomics server (http://www.phylogeny.fr) using the "One Click" mode with default settings. The numbers above the branches are tree supported values generated by PhyML using the approximate Likelihood Ratio (aLRT) statistical test. (**C**) Phylogenetic tree inferred from the inferred amino-acid sequence of HSP70 in different dipteran species. The tree was constructed using Phylogeny.fr tool at ExPASy Proteomics server (http://www.phylogeny.fr) using the "One Click" mode with default settings. The numbers above the branches are tree supported values generated by PhyML using the approximate Likelihood-Ratio (aLRT) statistical test.

Next, we wanted to understand if this transcript represents an additional isoform of the hsp70 gene or an as yet uncharacterized gene. To answer this question we extracted genomic DNA from *D. tonsa* larvae and probed using primers that include the additional sequence, 404 bp long. The amplification clearly showed two bands (Fig 2B). The shorter band was about 300 bp long while the longer band and fainter band was observed at about 700 bp, i.e. the difference in the length is exactly what we would expect in the case of two genes. This result suggests that these two transcripts are associated with two different genes in the *D. tonsa* genome. To further prove that this is the case, we calculated the relative intensity of the two bands, finding the *Dt-hsp70* to be exactly two-fold more intense than the alternative transcript (Fig 2C). This result suggests that two independent genes encode these transcripts. Moreover, these genes are

**Table 2. Summary of the characteristics of all sequences and their deduced proteins.**

| Isoform | Transcript | | | | | Protein | | |
|---|---|---|---|---|---|---|---|---|
| | Length (nt) | ORF (nt) | 5′UTR (nt) | 3′UTR (nt) | polyA | Length (aa) | tMW (kDa) | pI |
| *Dt-hsp70* | 2219 | 1904 | 166 | 167 | YES | 634 | 69.536 | 5.67 |
| *Dt-hsc70I* | 2273 | 1958 | 121 | 193 | YES | 652 | 71.451I | 5.47 |
| *Dt-hsc70II* | 2128 | 1532 | 410 | 185 | YES | 510 | 56.763 | 4.95 |

ORF = Open Reading Frame; UTR = Un-Translated Region; tMW = theoretical Molecular Weight.

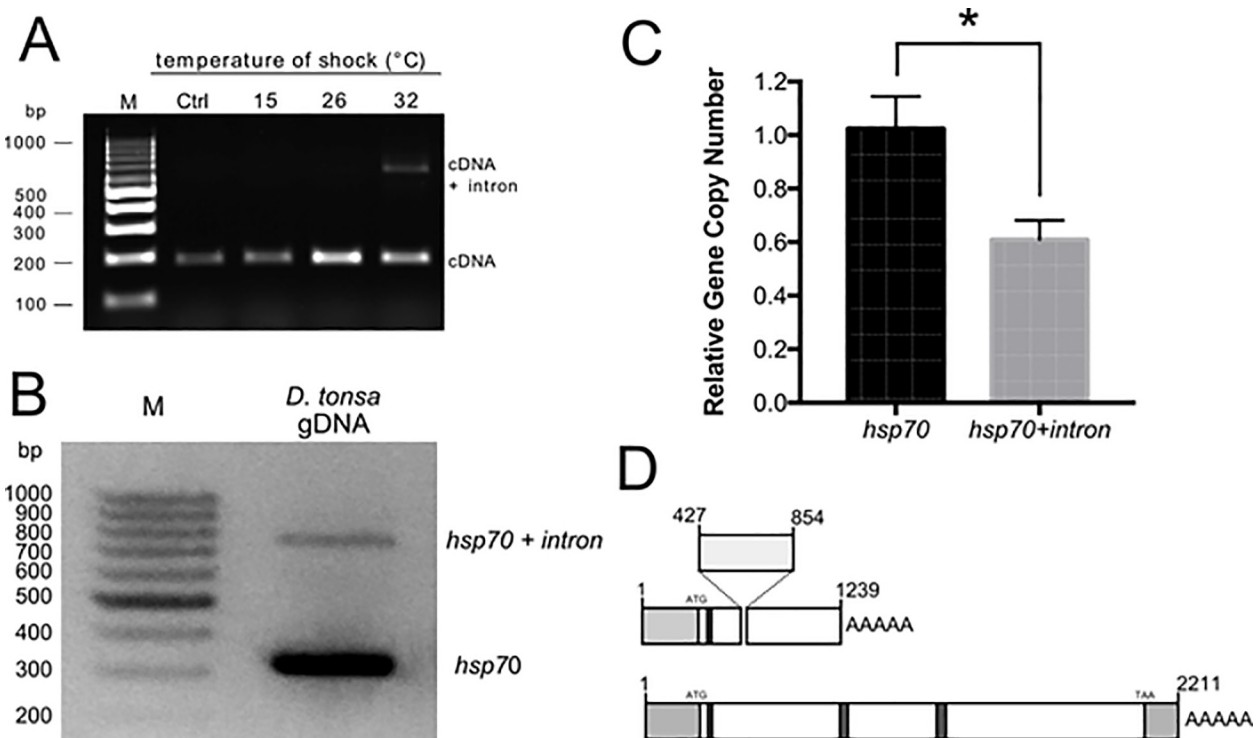

**Fig 2. Identification of an Hsp70 pseudogene in *Diamesa tonsa*.** (**A**) Agarose gel of *hsp70* PCR products from *D. tonsa* larvae control (Ctrl, 4˚C) or maintained for 1 h at 15, 26, and 32˚C. All PCR products are amplified from cDNA with primers for *hsp70* (**Table 1**). (**B**) Agarose gel electrophoresis of the PCR products amplified from *D. tonsa* genomic DNA with hsp70 sequence specific primers hsp70 F and hsp70 R (**Table 2**). (**C**) Relative gene copy number of hsp70 and hsp70 + intron assessed by Real-PCR analysis (n = 4) (Student *t*-test, * p ≤ 0.05). (**D**) Schematic representation of the two *hsp70* transcripts: light grey boxes are the 5′ and 3′ UTR, dark grey boxes indicated the position of the three characteristic HSP70 family domains.

represented at different copy number in the genome as demonstrated by the Gene Copy Number analysis by qPCR (Fig 2C).

Summarizing these results, we found that: i) in addition to the hsp70 gene, *D. tonsa* has a putative *hsp* pseudogene; ii) this gene encodes at least one transcript containing an insertion with respect to the *Dt-hsp70*; iii) the transcript has a partial overlap with the full length *Dt-hsp70*; iv) this transcript is likely a lncRNA. Given these characteristics, we named this gene *Dt-Ps-hsp70*.

## Multi-level analysis of changes in gene expression during thermal stress

To address the question of how *D. tonsa* adapts to thermal stress at the molecular level, we studied the gene expression changes in *Dt-hsp70*, *hsc70-I*, and *hsc70-II* induced by high temperature (15, 26, and 32˚C) in mature larvae. We hoped to set up the most complete experimental design to date for this organism and addressed the question by monitoring changes at the transcriptional, translational, and protein levels (Fig 3A). We extracted total RNA, polysome-associated RNA, and proteins from larvae exposed to 15, 26, and 32˚C. We studied all three levels of gene expression in the case of *Dt-hsp70* and the transcriptional and translational level for *Dt-hsp70-I* and *II*.

We found that *Dt-hsp70* and *Dt-hsc70-I* were significantly up regulated at 26 and 32˚C ($p < 0.001$), while a slight, but still statistically significant, decrease was observed for *Dt-*

*hsc70-II* at 26˚C (Fig 3B). The most dramatic transcriptional changes were observed for *Dt-hsp70*. A slight but significant positive change was observed in *Dt- hsc70-I* at 26 and 32˚C. *Dt-hsc70-II* did not change its recruitment on polysomes (Fig 3C). Interestingly *Dt-hsp70* showed strong variability among the four thermal stresses, with a decrease in mRNA recruitment at 15˚C ($p < 0.001$) and an increase at 32˚C ($p < 0.001$) (Fig 3C). Interestingly, at 26˚C, when the transcriptional up-regulation is at its maximal level (Fig 3B), no changes were observed at the polysomal (i.e. translational) level[37,38,39]. This result suggests that, despite robust transcriptional activation, the protein is likely not produced.

To test this hypothesis, we focused our attention on this transcript and studied the protein level by Western Blotting from total protein extracts (Fig 3D). GAPDH was used as protein loading control. HSP70 protein densitometry analysis clearly shows a trend which resembles the observed translational changes. Next, we compared the changes at all three levels (Fig 3E) and found that the fold-change profiles of protein and translational level were similar showing

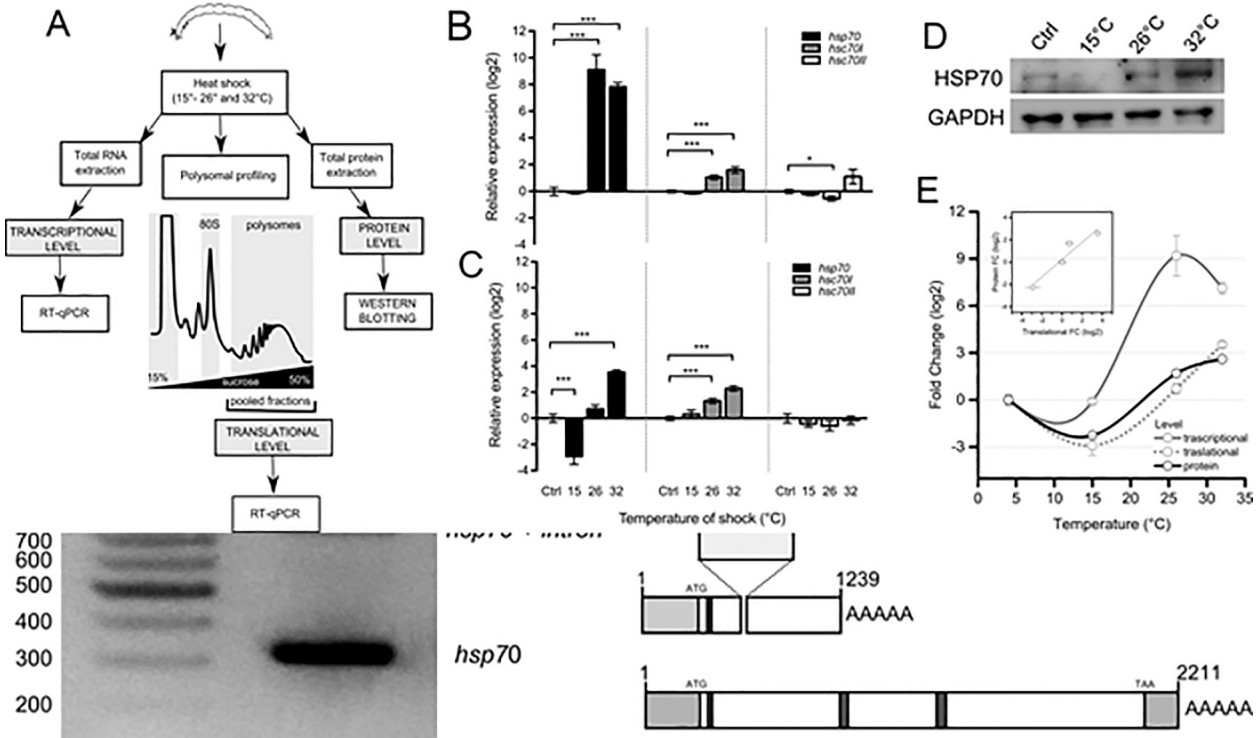

**Fig 3. Multi-level analysis of gene expression of *Dc-hsp70*, *hsc70-I* and *hsc70-II* during thermal stress.** (**A**) Experimental design for comparing changes in gene expression at multiple levels. Insects were exposed to thermal stress. Total RNA was extracted to analyse the changes at transcriptional level. In parallel, changes associated with mRNA recruitment to polysomes was obtained by RNA extraction from polysomal fractions. These were collected after polysomal profiling to assess the translational changes in gene expression. Finally, whole proteins were extracted to assess protein level. In the first two cases, all three transcripts were studied. For the protein level, only Hsp70 was monitored. (**B**) Transcriptional expression level for *hsp70*, *hsc70-I* and *hsc70-II*. Total RNA was extracted from *Diamesa tonsa* larvae control (Ctrl, 4˚C) or maintained for 1 h at 15, 26, and 32˚C. *hsp70*, *hsc70-I* and *hsc70-II* relative expression levels were measured by real-time PCR. *Actin* was used as housekeeping gene and the level of control (Ctrl, 4˚C) was set at 0. Error bars represent SE; n = 3 biological replicates and each assay was performed in triplicate. (**C**) Translational expression level for *hsp70*, *hsc70-I* and *hsc70-II*. Polysomal RNA was extracted from sucrose fractions corresponding to the polysomal peaks of larvae control (4˚C) or maintained for 1 h at 15, 26, and 32˚C. *hsp70*, *hsc70-I* and *hsc70-II* relative expression levels were measured by real-time PCR. *Actin* was used as housekeeping gene and the level of control (4˚C) was set at 0. (**D**) Western blot analysis of HSP70 protein level in larvae control (Ctrl, 4˚C) or maintained for 1 h at 15, 26, and 32˚C. GAPDH was used as a loading control. (**E**) Comparison of the log$_2$ Fold Change with respect to Ctrl of Transcriptional, Translational and Protein level after exposure to 15, 26 and 32˚C. Asterisks indicate statistically significant differences with respect to control (Student *t*-test, * $p \leq 0.05$, ** $p \leq 0.01$, *** $p \leq 0.001$). In the inset, the correlation between fold changes occurring at the translational and protein level was calculated ($R^2 = 0.922$). In the case of transcriptional and translational comparison, the correlation was ($R^2 = 0.389$), see S3A Fig.

very high correlation ($R^2$ = 0.922, Fig 3E, inset). Conversely, the profile of the transcriptional fold-change clearly differs, with no change at 15°C and remarkable increase at 26 and 32°C, and very low correlation with the translational level ($R^2$ = 0.389; S4A File).

## Role of Dt-Ps-hsp70 in translational control of HSP70 protein expression

Having shown that at 26°C of thermal stress there is a strong uncoupling of transcription and translation/protein level, we wondered whether, similar to observations in mammals during heat stress [29], the putative lncRNA *Dt-Ps-hsp70* transcript plays a role in attenuating the transcriptional boost.

To test this hypothesis, we studied the changes of *Dt-Ps-hsp70* at both the transcriptional and translational level by qPCR (Fig 4A, 4B), using conveniently designed primers (S4B Fig and Table 1). An increase in the expression of *Dt-Ps-hsp70* at transcriptional level was detected at 26°C and 32°C ($p < 0.001$) (Fig 4A), similar to observations of *Dt-hsp70* (Fig 3B).

This transcript was uploaded on polysomes, with a positive fold-change with respect to the control, exclusively at 26°C (Fig 4B). Next, we calculated the relative variations of translational and transcriptional changes (Translation Efficiency (ΔTE) in Fig 4C). A statistically significant increase in the ΔTE was observed in larvae stressed at 26°C ($p < 0.001$) and 32°C ($p < 0.05$). Strikingly, the most robust up-regulation was at 26°C, precisely when we observed the strongest uncoupling of transcription and translation in *Dt-Ps-hsp70* (Fig 4D).

Taken together these results suggest that the *Dt-Ps-hsp70* transcript likely competes with the *Dt-hsp70* transcript for ribosome recruitment, leading to attenuation of the global efficiency of HSP70 production at 26°C and suggesting that *Dt-Ps-hsp70* acts as a ribosome sponge to modulate the protein synthesis of HSP70.

## Discussion

The HSR is an important biochemical indicator for assessing levels of thermal stress and thermal tolerance limits stemming from the fact that protein conformation is a thermally sensitive weak-link [17]. Thus, species sensitivity inferred from HSR activation might be used as a proxy of their ecological valency [e.g., 23, 37, 38, 39] and their vulnerability to climate change.

Under HS, the cold adapted *D. tonsa* deploys a molecular strategy involving HSR (Fig 5). This strategy appears more complex than previously considered for larvae of *Diamesa* [13, 22] and other Diamesinae [22]. One *hsp70* gene (*Dt-hsp70*), one pseudogene *hsp70* (*Dt-Ps-hsp70*) and two isoforms of *hsc70* (*Dt-hsc70-I* and *Dt-hsc70-II*) have been sequenced in *D. tonsa*, experimentally showing differential expression under increased temperature (15, 26 and 32°C). This response was studied here at the transcriptional, translational, and protein level. As expected [22], *D. tonsa* showed strong activation of transcription of *hsp70* inducible forms (*Dt-hsp70*) at temperatures ≥ 26°C, and, to a minor extent, also of the cytoplasmic constitutive form (*Dt-hsc70-I*) at both 26 and 32°C. We found that *Dt-hsp70* and *Dt-hsc70-I* were significantly up-regulated at 26 and 32°C, whilst a slight, still statistically significant decrease was observed for *Dt- hsc70-II* at 26°C. In accordance with previous findings by other co-authors [40], the strongest transcriptional changes were observed for *Dt-hsp70*. Overall, *Dt-hsc70-I* cooperates with *Dt-hsp70* to help *D. tonsa* larvae survive stress.

The presence of more than one member of the *hsc70* family has been previously observed in other insects. For example, in the mosquito *Culex quinquefasciatus* [41] and the fruit fly *Drosophila melanogaster* [42] the hsp70 family includes seven heat shock cognate genes (*hsc70-1-7*). These genes are all expressed during normal growth, but the proteins show different subcellular localization: *hsc70-x* localizes to mitochondria, *hsc70-3* to the endoplasmic reticulum (ER) and the others either to the cytoplasm or the nucleus [43]. Phylogenetic analysis suggests

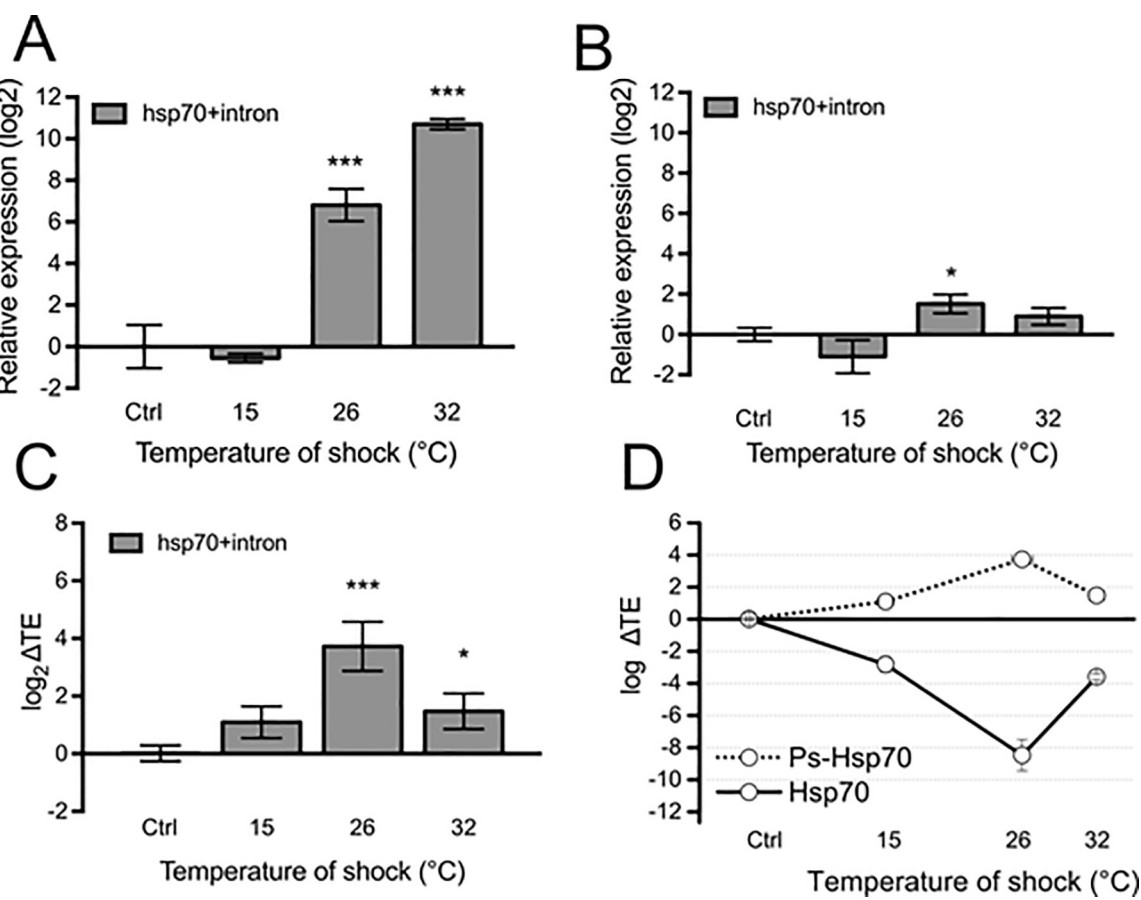

**Fig 4. Ps-Hsp70 is loaded on polysomes and acts as a putative ribosome sponge for Hsp70.** (**A**) Transcriptional expression level for *hsp70 + intron* mRNA. Total RNA was extracted from *Diamesa tonsa* larvae control (K, 4˚C) or maintained for 1 h at 15, 26, and 32˚C. Relative expression level was measured by real-time PCR. *Actin* was used as housekeeping gene and the level of control (Ctrl, 4˚C) was set at 0. Error bars represent SE; n = 3 biological replicates and each assay performed in triplicate. (**B**) Translational expression level of *hsp70 + intron*. Polysomal RNA was extracted from sucrose fractions corresponding to the polysomal peaks of larvae control (4˚C) or maintained for 1 h at 15, 26, and 32˚C. Relative expression level was measured by real-time PCR. *Actin* was used as housekeeping gene and the level of control (4˚C) was set at 0. (**C**) Translation Efficiency (log$_2$ ΔTE), calculated as the difference between the fold change at the polysomal level and the fold change at the sub-polysomal level, of *hsp70 + intron* in larvae control (Ctrl, 4˚C) or maintained for 1 h at 15, 26, and 32˚C. Asterisks indicate statistically significant differences in respect to the control (Student *t*-test, * $p \leq 0.05$, ** $p \leq 0.01$, *** $p \leq 0.001$). (**D**) Comparison between the Translation Efficiency (log$_2$ ΔTE) of Ps-Hsp70 and Hsp70. The ΔTE values for Hsp70 were obtained from data shown in Fig 3B and 3C.

that the cDNA sequences of *Dt-hsp70* and *Dt-hsc70-I* are more similar to those obtained from other Chironomidae than to other Diptera families. Conversely *Dt-hsc70-II* clusters with the *hsc70* of *Aedes aegypti* (Culicidae) and *hsc70-3* of *D. melanogaster* (Drosophilidae). This is probably due to the different subcellular localization of this *hsc70* member: *hsc70* of *Aedes aegypti* and *hsc70-3* of *D. melanogaster* correspond to proteins localized in the ER and both presented the Hsp70 C-terminal ER localization signal (KDEL) [43, 44] as does *Dt-hsc70-II*.

The discovery of a putative *hsp* pseudogene that encodes for at least one transcript containing an intron with respect to the *Dt-hsp70*, and its prediction as a lncRNA, is a very intriguing finding. *Dt-hsp70*, which is an inducible *Hsp70* gene, is intron-less as is typical in most organisms. As such, it does not require splicing for normal function [40, 45]. The lack of introns enables the transcribed mRNAs to move rapidly from the nucleus to the cytoplasm without splicing, significantly accelerating the HSR. Notably, splicing machinery is usually strongly

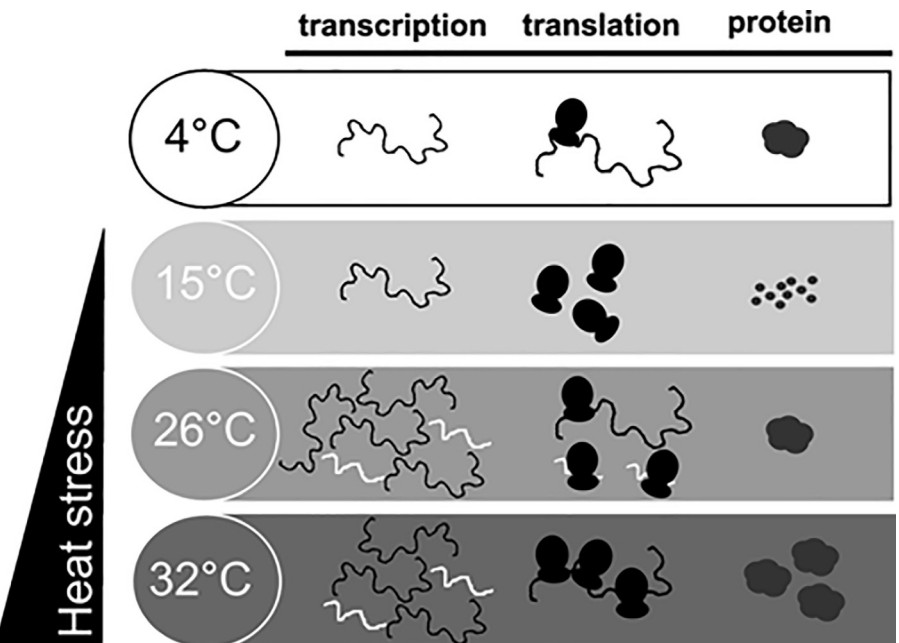

**Fig 5. Scheme summarizing the multi-level changes occurring during heat shock in *Diamesa tonsa*.**

inhibited by HS and other stressors. This fact may favour the selection of intron-less copies of *hsp* genes in the course of evolution [46, 47]. Introns have been considered to appear more frequently in cognate isoforms expressed at normal temperatures [48]. There are, however, several exceptions to this rule, such as in several species of fungi [49, 50], plants [51, 52], the nematode *Caenorhabditis elegans* [53], and in the *grasshopper Locusta migratoria* [54]. Interestingly, in these organisms, *hsp70* genes with introns were constitutively expressed [54]. Specifically, in *C. elegans*, with intron-exon arrangement of *hsp* genes, the splicing of *Hsp70* mRNA occurs with maximal efficiency after moderate HS while splicing of "normal" cellular genes, such as tubulin, is concomitantly blocked by a still unknown mechanism [55].

As has been reported for long noncoding RNAs of mammals [56], the *Dt-Ps-hsp70* transcript was uploaded on polysomes with a positive fold-change exclusively at 26˚C. Strikingly, the most robust up-regulation was at 26˚C, precisely when we observed the strong uncoupling of transcription and translation for the pseudogene. The fact that *Dt-Ps-hsp70* shares with *Dt-hsp70* the first half of the sequence leads us to propose a possible mechanism to explain the transcriptional/translational uncoupling and low protein production of HSP70 at 26˚C. Having the same 5′ sequence, it is reasonable that both transcripts have very similar probability of translation initiation events. This means that *Dt-Ps-hsp70* transcript can compete with the *Dt-hsp70* transcript for ribosome engagement. This condition can control the global efficiency of HSP70 production at 26˚C, suggesting that *Dt-Ps-hsp70* acts as a ribosome sponge and translational modulator of HSP70 protein levels. On the other hand, because of this function, we cannot exclude that the lncRNA is also localized in the nucleolus, but further experiments should be necessary to demonstrate it since we have performed our experiments of polysomal profiling using cytoplasmic nuclei and mitochondria-free lysates.

The uncoupling of transcription and translation can be explained by the consideration that, while important in facilitating tolerance of heat stress, the synthesis of HSPs has considerable metabolic costs, and it is activated only when essential for survival, otherwise it might be maladaptive [17]. Translation is known to drain more cellular energy than transcription [57]. The

synthesis of HSPs, and their function as ATP-consuming chaperones in protein folding reactions, can add considerably to the ATP demands of the cell, explaining the repression of basal transcription and translation [21]. Furthermore, our findings emphasize that transcription alone is not necessarily a reliable final readout of HSR in *D. tonsa*.

Consistent with our results, unspliced mRNAs are uploaded on polysomes for translation, resulting in the production of a pool of abnormal (truncated) Hsp70 proteins [58]. This finding emphasized that in this chironomid, as in many organisms from yeast [46, 49] and plants [52] to humans [59], exposure to heat does affect the spliceosome and hinders intron splicing.

The lncRNA discovered in *D. tonsa* is employed as a means of gene regulation and control of HSP70 synthesis under warming stress, as typically observed in higher eukaryotes, predicted by Jacob and Monod 58 years ago [60] and demonstrated in mammals [29].

Recently, great attention has been paid to lncRNAs that should not act as protein-coding transcripts but are still found associated with polysomes for largely unknown function [61, 62]. The literature suggests that the majority of the genomes of mammals and other complex organisms is in fact transcribed into lncRNAs (including those derived from introns), many of which are alternatively spliced and/or processed into smaller products [61]. At present, there are no clues about the involvement of lncRNAs in cold stenothermal organisms that are facing global warming challenges in the wild, such as *D. tonsa*. In this species, the gene encoding the lncRNA was preserved in the genome but is never expressed under natural conditions.

Our findings highlight for the first time the existence and the putative function of a lncRNA in HSR in a cold stenothermal insect. Why was an *hsp70* pseudogene positively selected by evolutionary driving forces? An attractive hypothesis is that the presence of inducible intron-isoforms could reflect an evolutionary strategy and adaptation to survive heat shock in cold adapted organisms, living constantly in cold waters [63]. Future experiments will be necessary to test this.

## Supporting information

**S1 Fig.** (**A**). Nucleotide and deduced amino acid sequence of *hsc70-I*. (**B**) Phylogenetic tree inferred from nucleotide sequences of hsp70 in different dipteran species. (**C**) Phylogenetic tree inferred from the inferred amino-acid sequence of HSP70 in different dipteran species.
(DOCX)

**S2 Fig. Sequence of the *Dt-hsc70-II* mRNA constitutive isoform.**
(DOCX)

**S3 Fig. Alignment of the *Dt-hsp70* and the pseudogene transcripts.**
(DOCX)

**S4 Fig.** (**A**) Correlation between fold change at translational and transcriptional level. (**B**) Schematic representation of the two hsp70 transcripts.
(DOCX)

**S1 Raw Images. Original blot and gel data.**
(DOCX)

## Acknowledgments

We thank Alessandra Franceschini and Francesca Paoli (MUSE-Museo delle Scienze of Trento) for animal collection.

## Author Contributions

**Conceptualization:** Valeria Lencioni.

**Data curation:** Paola Bernabò, Valeria Lencioni.

**Formal analysis:** Paola Bernabò.

**Funding acquisition:** Valeria Lencioni.

**Investigation:** Paola Bernabò.

**Methodology:** Paola Bernabò, Gabriella Viero.

**Project administration:** Valeria Lencioni.

**Resources:** Valeria Lencioni.

**Supervision:** Gabriella Viero, Valeria Lencioni.

**Validation:** Gabriella Viero, Valeria Lencioni.

**Visualization:** Gabriella Viero.

**Writing – original draft:** Paola Bernabò.

**Writing – review & editing:** Gabriella Viero, Valeria Lencioni.

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
