## [Decision Letter · Decision Letter 0]

13 Feb 2020

PONE-D-19-34496

A long noncoding RNA acts as a post-transcriptional regulator of heat shock protein 70 kDa synthesis in the cold hardy Diamesa tonsa under heat shock

PLOS ONE

Dear Dr.Lencioni,

Thank you for submitting your manuscript to PLOS ONE. Your article has been reviewed by two reviewers.  Both the reviewers felt that the manuscript reports interesting data, but both of them suggested improvements to make it eligible for publication.  Therefore, we invite you to submit a revised version of the manuscript that addresses the points raised during the review process.

We would appreciate receiving your revised manuscript by Mar 29 2020 11:59PM. To enhance the reproducibility of your results, we recommend that if applicable you deposit your laboratory protocols in protocols.io, where a protocol can be assigned its own identifier (DOI) such that it can be cited independently in the future. For instructions see: http://journals.plos.org/plosone/s/submission-guidelines#loc-laboratory-protocols

We look forward to receiving your revised manuscript.

Kind regards,

Swati Palit Deb

Academic Editor

PLOS ONE

Journal Requirements:

1. Thank you for inlcuding your funding statement; "This work was partially supported by the Cassa di Risparmio di Trento e Rovereto Foundation within the RACE-TN Project (“Valutazione del rischio ambientale dei contaminanti emergenti nei fiumi trentini: effetti sulla vita selvatica e sull’uomo”/“Environmental Risk assessment of emerging contaminants in Trentino rivers: effects on wildlife and human health”, Grant CARITRO n. 2015.0199; October 2015-January 2018). The funder had no role in study design, data collection and analysis, decision to publish, or preparation of the manuscript. The funder supported 1 m/m of the contract of the first author (Paola Bernabò) and the purchase of some consumables."

Reviewers' comments:

Reviewer's Responses to Questions

**Comments to the Author**

1. Is the manuscript technically sound, and do the data support the conclusions?

Reviewer #1: Yes

Reviewer #2: Partly

2. Has the statistical analysis been performed appropriately and rigorously? 

Reviewer #1: Yes

Reviewer #2: Yes

3. Have the authors made all data underlying the findings in their manuscript fully available?

Reviewer #1: Yes

Reviewer #2: Yes

4. Is the manuscript presented in an intelligible fashion and written in standard English?

Reviewer #1: Yes

Reviewer #2: Yes

5. Review Comments to the Author

Reviewer #1: In this manuscript, the authors show how a stenothermal organism controls gene expression at the transcriptional, translational, and protein levels. They identify a novel pseudogene hsp70, encoding a putative lncRNA that exhibits a post-transcriptional control on HSP70 protein levels. The authors have carried out an extensive study to show that pseudogenes, indeed, are important regulators of gene regulation and should not be considered as junk DNA. In this context, the authors are requested to include this recent review in their references:

https://www.ncbi.nlm.nih.gov/pubmed/31848477

All the experiments have been carried out meticulously with the proper controls and statistical analysis. However, a couple of questions need to be addressed and a few minor changes.

1. I was wondering if the pseugodene encoded lncRNA can be knocked out using siRNA? If so, what effect would that have on the post-transcriptional control of HSP70? Have the authors tried checking this? Alternatively, they could try and include a couple of sentences in the discussion section.

2. The authors have looked at the gene expression in the polysome fractions. Have they looked at the protein expression of the hsp proteins in the polysome fractions?

3. Optional: Could the lncRNA be specifically localizing to the nucleolus especially since these are centers of ribosome production and since the authors show that Dt-Ps-hsp70 acts as a ribosome sponge to modulate the protein synthesis of HSP70. Maybe the authors could include a couple of sentences to discuss this, if possible.

Minor:

Title: Remove "kDa" and include the abbreviated form in brackets (e.e.g HSP70)

Page 5...Line 100...Replace "cox1" with "COX1"

Page 12...Line 274...Replace "long nc-RNAs" with "lncRNAs".

Page 16...Line 366...Is hsc70-x one of the cognate hsc genes or is it a typo?

Reviewer #2: In this manuscript the authors have proposed existence of a long noncoding RNA as post-transcriptional regulator of HSP 70 in Diamesa tonsa under heat shock. While authors provide compelling evidence of pseudo-hsp 70 modulating heat thermal stress response, more functional analysis is needed to prove the correlation. Have the authors tried genome editing tools to maybe try to silence the ps-hsp 70 and then examine its effect on hsp70? Or maybe even try to overexpress it.

6. PLOS authors have the option to publish the peer review history of their article (what does this mean?). If published, this will include your full peer review and any attached files.

Reviewer #1: No

Reviewer #2: No

---

## [Author Response · Author response to Decision Letter 0]

26 Feb 2020

Reviewers' Comments to Author and Replies by the Authors to Reviewers:

Reviewer #1: 

In this manuscript, the authors show how a stenothermal organism controls gene expression at the transcriptional, translational, and protein levels. They identify a novel pseudogene hsp70, encoding a putative lncRNA that exhibits a post-transcriptional control on HSP70 protein levels. The authors have carried out an extensive study to show that pseudogenes, indeed, are important regulators of gene regulation and should not be considered as junk DNA. In this context, the authors are requested to include this recent review in their references: https://www.ncbi.nlm.nih.gov/pubmed/31848477

R: We thank the Reviewer for his/her suggestion to include this important review. We now included the citation as requested.

All the experiments have been carried out meticulously with the proper controls and statistical analysis. However, a couple of questions need to be addressed and a few minor changes.

1. I was wondering if the pseugodene encoded lncRNA can be knocked out using siRNA? If so, what effect would that have on the post-transcriptional control of HSP70? Have the authors tried checking this? Alternatively, they could try and include a couple of sentences in the discussion section.

R: We definitively agree with the Reviewer that using genetic tools to further prove and understand the function of the lncRNA would be very intriguing and relevant. Working with a wild species is of clear importance, but at the same has a number of limitations. In fact, wild organisms cannot be easily reared under laboratory conditions and no genetic manipulation procedures are yet available and established. In addition, no cell lines for this species are available, hampering the application of any silencing approach in our biological system. 

2. The authors have looked at the gene expression in the polysome fractions. Have they looked at the protein expression of the hsp proteins in the polysome fractions?

R: The Reviewer’s question is interesting but can have different interpretations. If the referee was asking for the de-novo protein production, a polysomal profiling is not a reliable technique. It is known that Hsps proteins do indeed interact with ribosomes, thus, if the Reviewer was asking for changes in the interaction of Hsp’s to polysome, it is very likely that the interaction is deeply affected and driven by the mass effect of changes in overall protein level. 

3. Optional: Could the lncRNA be specifically localizing to the nucleolus especially since these are centers of ribosome production and since the authors show that Dt-Ps-hsp70 acts as a ribosome sponge to modulate the protein synthesis of HSP70. Maybe the authors could include a couple of sentences to discuss this, if possible.

R: The Reviewer is right and it is likely that the lncRNA is also localized in the nucleolus. All our experiments of polysomal profiling were performed using cytoplasmic nuclei and mitochondria-free lysates. Therefore, the localization of the lncRNA with polysomes is due only to the cytoplasmic lncRNA. We cannot exclude that the lncRNA, if also localized in the nucleus, could exert additional roles in the nucleolus; further experiments are necessary to state it, but this go beyond the aim of this paper.

As suggested, we mentioned this additional possibility in the new version of the manuscript.

Minor:

Title: Remove "kDa" and include the abbreviated form in brackets (e.e.g HSP70)

R: (HSP70) was added and kDa removed.

Page 5...Line 100...Replace "cox1" with "COX1"

R: Done.

Page 12...Line 274...Replace "long nc-RNAs" with "lncRNAs".

R: Done.

Page 16...Line 366...Is hsc70-x one of the cognate hsc genes or is it a typo?

R: hsc70-x is a typo. 

Reviewer #2: 

In this manuscript the authors have proposed existence of a long noncoding RNA as post-transcriptional regulator of HSP 70 in Diamesa tonsa under heat shock. While authors provide compelling evidence of pseudo-hsp 70 modulating heat thermal stress response, more functional analysis is needed to prove the correlation. Have the authors tried genome editing tools to maybe try to silence the ps-hsp 70 and then examine its effect on hsp70? Or maybe even try to overexpress it.

R: The Reviewer raised a very important and critical point. Using genetic tools to further prove and understand the function of the lncRNA would be very intriguing and relevant. Despite of clear importance, these procedures cannot be applied reliably to wild species, working with whom has a number of limitations. In fact, wild organisms cannot be easily reared under laboratory conditions and no genetic manipulation procedures are yet available and established. In addition, no cell lines for this species are available, hampering the application of any silencing approach in our biological system.

---

## [Editor Report · Decision Letter 1]

2 Mar 2020

A long noncoding RNA acts as a post-transcriptional regulator of heat shock protein (HSP70) synthesis in the cold hardy Diamesa tonsa under heat shock

PONE-D-19-34496R1

Dear Dr. Lencioni,

We are pleased to inform you that your manuscript has been judged scientifically suitable for publication and will be formally accepted for publication once it complies with all outstanding technical requirements.

With kind regards,

Swati Palit Deb

Academic Editor

PLOS ONE
---

## [Editor Report · Acceptance letter]

5 Mar 2020

PONE-D-19-34496R1 

A long noncoding RNA acts as a post-transcriptional regulator of heat shock protein (HSP70) synthesis in the cold hardy *Diamesa tonsa* under heat shock 

Dear Dr. Lencioni:

I am pleased to inform you that your manuscript has been deemed suitable for publication in PLOS ONE. Congratulations! Your manuscript is now with our production department. 

With kind regards,

on behalf of

Dr. Swati Palit Deb 

Academic Editor

PLOS ONE